# VIP: Vision Instructed Pre-training for Robotic Manipulation

Zhuoling Li[1]  Liangliang Ren[2]  Jinrong Yang[‡, 2 3]  Yong Zhao[2]  Xiaoyang Wu[1]  Zhenhua Xu[4]
Xiang Bai[5]  Hengshuang Zhao[1]

https://lizhuoling.github.io/VIRT_webpage/

## Abstract

The effectiveness of scaling up training data in robotic manipulation is still limited. A primary challenge in manipulation is the tasks are diverse, and the trained policy would be confused if the task targets are not specified clearly. Existing works primarily rely on text instruction to describe targets. However, we reveal that current robotic data cannot train policies to understand text instruction effectively, and vision is much more comprehensible. Therefore, we introduce utilizing vision instruction to specify targets. A straightforward implementation is training a policy to predict the intermediate actions linking the current observation and a future image. Nevertheless, a single future image does not describe the task target in insufficient detail. To handle this problem, we propose to use sparse point flows to provide more detailed information. Extensive tasks are designed based on real and simulated environments to evaluate the effectiveness of our vision instructed pre-training (VIP) method. The results indicate VIP improves the performance on diverse tasks significantly, and the derived policy can complete competitive tasks like "opening the lid of a tightly sealed bottle".

## 1. Introduction

Following the data scaling law, the natural language processing and computer vision communities have achieved remarkable breakthroughs (Achiam et al., 2023; Liu et al., 2024). In the robotics community, there are also many attempts of incorporating more robotic data (O'Neill et al., 2024; Kim et al., 2024). However, the effectiveness of large-scale training in robotic manipulation is still limited (Billard

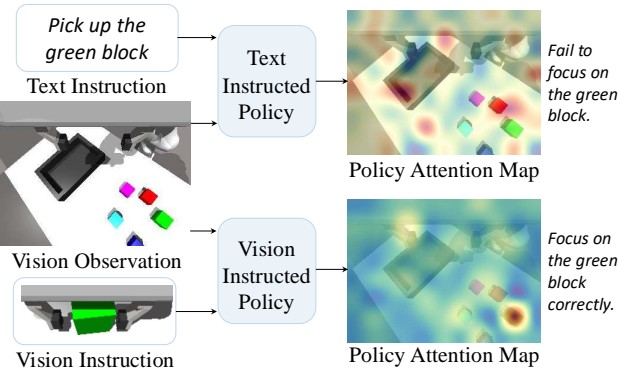

Figure 1. Visualization comparison between the action attention maps of the text instructed policy and vision instructed policy. We can observe that the text instructed policy is confused about which region to concentrate on, while the vision instructed policy focuses on the target correctly. This phenomenon suggests that vision instruction is more comprehensible by policy networks.

& Kragic, 2019). This is partly because manipulation data is ambiguous due to its multi-modal nature (Chi et al., 2023). Given the same observation, there could exist multiple objects to manipulate and the potential actions are diverse, such as picking, sorting, and rotating.

To handle this ambiguity, it is important to describe task targets clearly to the policy. A common practice in existing manipulation pre-training paradigms is describing task targets using text instructions like "pick up the green block". These paradigms expect that the trained policy understands what the green block is in the input image and predicts the action sequence of picking it up. Therefore, the policy needs to align the information between text and vision, such as the words "green block" and the appearance of this green block. However, we find that existing manipulation data is not diverse sufficiently to train a policy to own this capability, which demands millions of image-text pairs as suggested in previous literature (Liu et al., 2024). As shown in Fig. 1, the text instructed policy fails to concentrate on the green block specified in the text instruction, and the robot hand often grasps a block in a false color.

In this work, we reveal that vision instruction, such as a future image predicting the grasp moment, is much more comprehensible by policies in the current robotic data di-

[1]HKU [2]CVTE [3]SYSU [4]THU [5]HUST. ‡Project leader. Correspondence to: Hengshuang Zhao <hszhao@cs.hku.h>.

*Proceedings of the 42st International Conference on Machine Learning*, Vancouver, Canada. PMLR 267, 2025. Copyright 2025 by the author(s).

versity scale. As illustrated in Fig. 1, the vision instructed policy focuses on the correct block and grasps it successfully. This is because both the observation and instruction lie in the vision domain, avoiding the need for numerous data to bridge the feature gap between different domains. We can also understand this phenomenon from the view of analyzing human infant behaviors, as existing robotic policies are built upon infant-level intelligent networks like ResNet (He et al., 2016) due to real-time deployment requirements. When we tell an infant to pick up a green block, the infant cannot understand the instruction even after many iterations of text explanations. However, if we show the infant what is the green block and how to grasp it through vision, the infant could gradually make correct reactions.

Therefore, vision instruction is more suitable for prompting policies in the current diversity scale of robotic data. A natural idea of using vision instruction is feeding the policy with future images besides the current observation, and the policy is optimized to predict correct actions that make the robot reach the status described in future images. However, a single future image is insufficient to describe manipulation dynamics and more frames lead to dramatically increasing computation cost. To specify the manipulation procedures clearly while maintaining an acceptable computational burden, we propose to represent the intermediate action information with sparse point flows. Specifically, we sample sparse points in the current vision observation and track the moving dynamics of these points. Therefore, the input to the pre-trained policy in VIP includes the current image, a future image describing the target, and the sparse point flows between the current and future images.

However, sparse point flows and future images are unavailable during inference. To handle the lack of sparse point flows, we progressively remove them during pre-training by random masking. This design gradually boosts the action prediction challenge and helps the policy learn more meaningful representation (Oquab et al., 2024). For the future image in inference, we replace it as the cropped region of the object to manipulate from the current observation. We find this design achieves better performance, as it excludes the disturbance from image background while also clearly specifying the target object. Additionally, this design enables us to specify the object manipulation order dynamically.

Combining the above designs, **V**ision **I**nstructed **P**re-training (VIP) is derived. Based on VIP, we pre-train our designed fully Transformer-based policy using 1.7B of manipulation data (Khazatsky et al., 2024). Extensive real-robot and simulated tasks are designed to verify the performance. For real-robot experiments, we devise tasks to validate different capabilities of policies from various perspectives, and the tasks include pouring blueberries into a juicer cup, opening the lid of a tightly sealed bottle, and cleaning plates

given the online specified order. In simulated experiments, we build a real-time human hand pose acquisition system to teleoperate robotic hands in Isaac Gym (Makoviychuk et al., 2021) and design several tasks of transporting and stacking color blocks following task instructions. The experimental results demonstrate the superior performance of VIP.

## 2. Related Work

**Demonstration Learning based Robotic Manipulation**. Robotic manipulation has advanced markedly due to the integration of machine learning techniques (Fang et al., 2019). Among existing methodologies, demonstration learning has garnered significant attention for its training efficiency (Zhao et al., 2023). The fundamental premise of demonstration learning is that a human teacher performs a task while the robot records the relevant data, such as sensory inputs, actions, and corresponding outcomes. This recorded data is subsequently used to train models that allow the robot to replicate the demonstrated behavior in similar situations (Florence et al., 2022).

Existing demonstration learning policies can be broadly categorized into two groups, *i.e.*, explicit policies (Fu et al., 2024b) and implicit policies (Chi et al., 2023). Among them, explicit policies directly map environment observations to actions, and the policy output is supervised with human demonstration trajectories by computing regression losses (Fu et al., 2024a). By contrast, implicit policies define the distributions of actions with energy-based models, where predicting the next action is framed as identifying the manipulation trajectory with minimal energy (Chi et al., 2024). This modeling approach allows for the natural representation of multi-modal distributions in manipulation trajectories. Consequently, some studies suggest implicit policies are more advantageous for robotic manipulation learning (Florence et al., 2022). Nevertheless, we contend that explicit policies offer faster response speeds due to their simplicity, which is crucial for robotic manipulation. In addition, the iterative decoding mechanism inherent in Transformer models is similar to the denoising process in implicit policies, and thus can also handle the multi-modal ambiguity in robotic manipulation to some extent. Hence, in this work, we develop a fully Transformer-based policy adopting the explicit prediction paradigm.

**Robotic Pre-training**. Recent advancements in natural language processing and computer vision demonstrate the efficacy of first pre-training models on large-scale data and then fine-tuning them for specific downstream applications (Achiam et al., 2023; Wang et al., 2023b). Drawing inspirations from these successes, the robotic learning community begins to explore pre-training paradigms to enhance robotic manipulation capabilities (Radosavovic et al., 2023; Nair et al., 2023; Majumdar et al., 2023). The principal idea be-

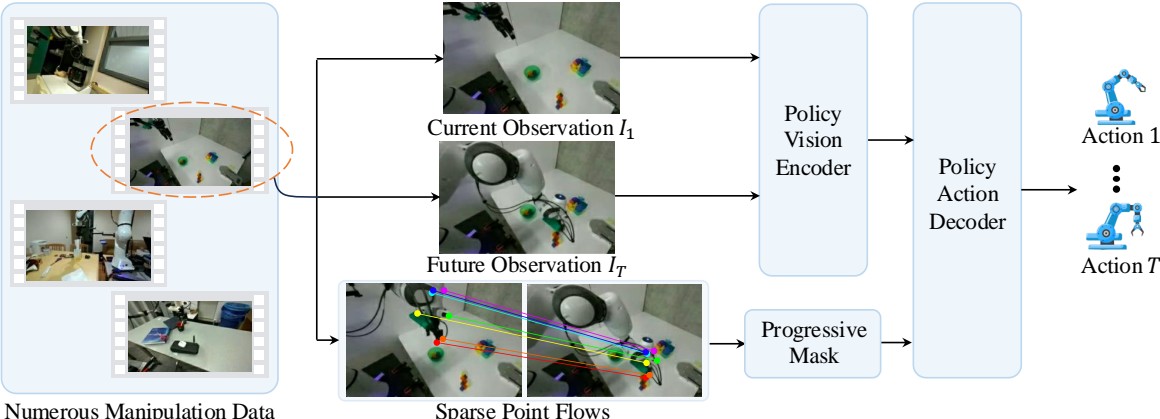

*Figure 2.* Overall pipeline of VIP. The input to the pre-trained policy includes two image frames (the observation frame and future frame) and sparse point flows, which describe the changing dynamics of the scene. The sparse point flows are gradually removed by the progressive mask module during pre-training.

hind robotic pre-training is to first expose the robotic policy to a wide range of tasks and environments, allowing it to learn generalizable representations for diverse robotic tasks (Brohan et al., 2022). Subsequently, a fine-tuning phase on specific manipulation tasks is conducted, utilizing the previously gained prior knowledge to enhance performance and efficiency (Zitkovich et al., 2023).

Pre-training a policy requires a substantial amount of data (Fang et al., 2020). However, robotic manipulation data is expensive to collect. To mitigate this issue, some methods generate data through simulated environments (Wang et al., 2023a). However, significant discrepancies in appearance and motion dynamics between simulated and real-robot data limit thier effectiveness. Several methods employ large language models to generate grasp positions for objects in 2D images (Vuong et al., 2024), but this approach is constrained by its two-dimensional output, whereas robotic manipulation occurs in a three-dimensional space. Recently, collaborative efforts among various institutions have led to the creation of large-scale datasets by merging existing data sources (O'Neill et al., 2024) or collecting new data across diverse scenarios (Khazatsky et al., 2024). Thanks to these datasets, a handful of promising pre-trained policies are derived (Kim et al., 2024). Nevertheless, robotic manipulation trajectories remain ambiguous if without appropriate task instructions as prompt. Existing pre-training algorithms predominantly use text instructions to inform the policy, which restricts the pre-training effectiveness, as previously noted.

**Robotic Instruction**. Robotic manipulation learning is intrinsically a long-sequence auto-regressive problem, often involving thousands of action steps within a short duration (Chen et al., 2024). Therefore, a basic challenge in manipulation lies in the ability of a policy to determine the appropriate actions based on the current observation. This problem is especially serious if the task to perform

involves multi-object manipulation or there are many potential operation steps (Shi et al., 2023). To alleviate this problem, instructions are demanded to guide policies with task-specific information. In previous works, the instructions are mostly represented as natural language (Brohan et al., 2022; Zitkovich et al., 2023). Alternatively, some researchers have also explored voice instructions (Shi et al., 2024), but voice information similarly poses learning difficulties. By contrast, images are more readily comprehensible for policy networks, as the commonly adopted backbones of these networks are already pre-trained on extensive image datasets (He et al., 2016). This pre-existing visual understanding in robotic policies is akin to the innate vision comprehension in human infants. Despite the potential of visual instructions, their applications remain unexplored in the context of robotic manipulation. Existing studies on visual instructions have primarily focused on goal images within game-based reinforcement learning (Yuan et al., 2024) and navigation (Majumdar et al., 2022). This work aims to bridge the gap in exploring visual instructions for robotic manipulation.

## 3. Method

### 3.1. Vision Intructed Pre-training

A data sample for robotic manipulation pre-training consists of two parts, a video sequence $V = \{I_1, I_2, \cdots, I_T\}$ and the corresponding robot actions $A = \{\bar{a}_1, \bar{a}_2, \cdots, \bar{a}_T\}$. In a real robotic manipulation application, future information is unavailable, and a policy $\pi$ needs to predict future actions $A$ using only the current observation $I_1$. However, manipulation trajectories are highly diverse. Directly regressing $A$ based on $I_1$ is too ambiguous for $\pi$. To address this problem, instruction is demanded in pre-training to disclose future information to $\pi$ to guide the generation of future actions.

The previous robotic pre-training paradigms often disclose

future manipulation trajectory information with text-based task description. These methods take vision observation and text description as input to predict actions, the process of which demands aligning the information among three domains (vision, text, and action). Nevertheless, we reveal that existing robotic data is not diverse enough to align text and vision representation. In addition, it is challenging to elaborate manipulation procedures clearly based on text description. Different from previous paradigms, this work proposes VIP, which pre-trains a policy $\pi$ to predict actions based on solely vision information. As illustrated in Fig. 2, the vision information includes current observation $I_1$, future observation $I_T$, and sparse point flows. Notably, each observation could contain multiple images captured by different cameras at the same timestamp.

In VIP, we first transform $I_1$ and $I_t$ as visual features $F_1$ and $F_t$ by a shared encoder like ResNet (He et al., 2016) in the pre-trained policy. Besides, we utilize sparse point flows in moving image regions to describe the missing intermediate robot manipulation information. We gradually remove the point flows through the progressive mask module shown in Fig. 2, and the remaining flows are transformed into the feature $F_p$ by a simple MLP layer. Refer to Section 3.2 for details about how the point flows are generated and masked. According to Fig. 2, $F_1$ is the environment observation. $F_T$ and $F_p$ are the vision instruction, which describes the future moving dynamics. $F_1$, $F_T$, and $F_p$ are input to the action decoder (*e.g.*, Transformer decoders or diffusion heads) of the pre-trained policy to produce $T$ action predictions $\{a_t\}_{t=1}^{T}$ and corresponding Laplacian uncertainty values $\{\sigma_t\}_{t=1}^{T}$. We optimize the pre-trained policy by minimizing a loss $L$ constructed based on $\{a_t\}_{t=1}^{T}$, $\{\sigma_t\}_{t=1}^{T}$, and $\{\bar{a}_t\}_{t=1}^{T}$ as:

$$L = \frac{1}{T} \sum_{t=1}^{T} \left( \frac{\sqrt{2}|a_t - \bar{a}_t|}{\sigma_t} + \log \sigma_t \right). \quad (1)$$

Notably, $\{\sigma_t\}_{t=1}^{T}$ has no ground truth and is learned in an unsupervised manner. Refer to previous literature (Li et al., 2022; Kendall & Gal, 2017) for why uncertainty can be captured in this way. According to Eq. (1), we can observe that the learned uncertainties $\{\sigma_t\}_{t=1}^{T}$ exhibit higher values for more ambiguous action segments. Consequently, larger $\{\sigma_t\}_{t=1}^{T}$ result in a smaller penalization for the discrepancies between predicted actions $\{a_t\}_{t=1}^{T}$ and demonstrated actions $\{\bar{a}_t\}_{t=1}^{T}$. This property enables $\pi$ to concentrate on more deterministic action segments.

## 3.2. Sparse Point Flow

Consecutive frames in a video contain numerous redundant information for recording the changing dynamics of a robot. Therefore, employing a video sequence to describe manipulation procedures to a pre-trained policy leads to huge computation cost. Differently, as depicted in Fig. 3, using

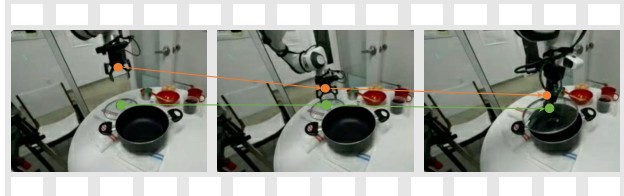

*Figure 3.* The conceptual diagram of sparse point flow. Consecutive frames in a video comprise numerous pixels and contain much redundant information for describing the movement of a robot hand. By contrast, a small group of points tracking moving pixels, namely sparse point flows, are much more efficient.

sparse point flows in moving image regions is much more efficient. Compared with using an image sequence where every frame comprises tens of thousands of pixels, the sparse point flows only have tens of points and can describe the robot hand manipulation process clearly.

In this work, we generate sparse point flows based on Co-Tracker (Karaev et al., 2024), a state-of-the-art point tracker. A simple implementation of using CoTracker is first randomly sample some points in the first frame of a video sequence and then provide these points to CoTracker. Co-Tracker can automatically track the point flows in subsequent frames. However, this implementation suffers from two drawbacks: (i) We only concern the changing regions in the frames of a video, while this implementation wastes many points in the static background. (ii) In many frames, the robot hand is invisible or not moving. These frames should not be incorporated into the pre-training data.

To handle the aforementioned two drawbacks, we adopt a two-stage strategy to generate the sparse point flows. Specifically, we first uniformly sample points in the first frame and produce point flows using CoTracker. Then, we remove the point flows that keep static among frames. If the remaining moving flows are too few (*e.g.*, fewer than three), we do not add this video sequence into pre-training data. Conversely, if there are enough moving flows, we randomly sample more points around these remaining moving flows to produce more sufficient point flows. After these two phases of point sampling, the derived sparse point flows describe the changing dynamics of the video sequence in a both efficient and detailed manner.

Nevertheless, we find that too abundant point flows prevent the policy from learning very meaningful vision representation. There are mainly two problems. On the one hand, infering robot hand actions from point flows is easier than using vision observations. Therefore, very sufficient point flow information makes the policy become lazy to explore the feature in vision observations. On the other hand, the point flows are unavailable in inference, so adopting point flows during pre-training causes an input information gap between pre-training and inference. To address these two prob-

lems, we gradually remove point flows during pre-training by masking them with an increasing probability. Specifically, assuming there are totally $N$ pre-training iterations, we randomly mask $\alpha_n$ of point flows at the $n_{\text{th}}$ iteration, where $\alpha_n = \min(1.25 \times \frac{n}{N}, 1)$. In this way, the point flow is completely removed at the end of pre-training, and thus the above two problems are tackled.

### 3.3. Vision Instruction after Pre-train

As the robots used in applications are often different from the robots for gathering pre-train data, a small handful of demonstration data is needed to fine-tune the pre-trained policy. To maintain usage consistency, The implementation of the policy in fine-tuning is the same as the one in inference. Therefore, we elaborate them together in this part.

The main difference between pre-training and the subsequent fine-tuning and inference phases is the vision instruction. As shown in Fig. 2, the vision instruction in pre-training consists of two parts, the future frame and sparse point flows. As introduced in Section 3.2, the sparse point flows are completely removed at the end of pre-training, so the problem is how to get the future frame. To address this problem, we first try training a diffusion based world model (Parmar et al., 2024; Ho et al., 2020) to predict the future image. It can be observed from Fig. 4 that the world model predicted image is more accurate in simple simulated scenarios but not satisfactory in real scenarios. For example, as shown in the second row of Fig. 4, the world model output is easy to collapse to being the same as the input image.

There are mainly two reasons resulting in this phenomenon. First of all, the future observation $o_{t+1}$ is affected by both the current state $s_t$ and action $a_t$. However, the input to the world model only includes $s_t$, because $a_t$ is unavailable. Why we predict $o_{t+1}$ is to use $o_{t+1}$ for guiding the generation of $a_t$. In other words, predicting $o_{t+1}$ demands $a_t$ and outputting $a_t$ depends on $o_{t+1}$, which is an endless loop. Hence, $a_t$ is not input to the world model. This restricts the future image prediction accuracy. Secondly, compared with simulated environments, the object interaction dynamics like collision in real scenarios is more complex.

According to the above analysis, we cannot employ a future frame as the vision instruction. To bridge this gap, we propose to replace the future frame in pre-training as the cropped image region of the object to manipulate during fine-tuning and inference. Although different from future frames, the cropped image also specifies where the robot hand should move towards for manipulation, as illustrated in the third row of Fig. 4. Our experimental results indicate that the pre-trained policy can adapt to using this new vision instruction based on only a little fine-tuning data. In addition, as the cropped image delivers which is the next object to manipulate, users can change the object manipulation

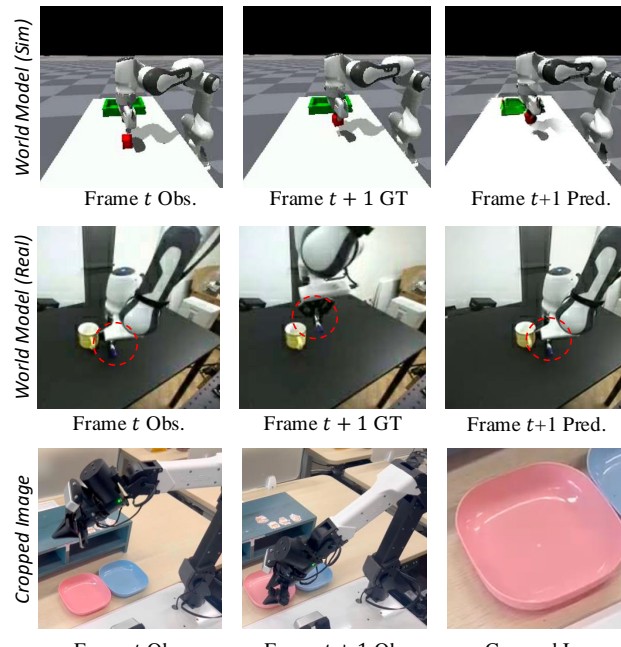

*Figure 4.* Visualization of different vision instructions. The three columns of images in the first and second rows show the world model input, future ground truth, and future image prediction in simulated and real scenarios. The third row illustrates the cropped image of the object to manipulate.

order dynamically by providing different cropped image regions during inference. The cropped image regions can be specified in various ways, *e.g.*, user eye gaze recognization, user instruction understanding based on an LLM, an external detection or segmentation model, etc.

### 3.4. Vision Instructed Robotic Transformer

To fully exploit the potential of VIP, we develop a fully Transformer-based policy named **V**ision **I**nstructed **R**obotic **T**ransformer (VIRT). In VIRT, we implement the policy vision encoder in Fig. 2 with twelve Transformer encoders. These encoders are initialized from the weight of DINOv2 (Oquab et al., 2024). Before inputting $I_1$ and $I_t$ to the encoders, we randomly mask a ratio $\tau$ of their pixels. This operation forces VIRT to be more sensitive to image details, as it can only depend on random remained image pixels to predict actions. The policy action decoder in Fig. 2 is modeled as three Transformer decoders. $T$ learnable queries are randomly initialized to interact with $F_1$, $F_t$, and $F_p$ in decoders to produce $\{a_t\}_{t=1}^{T}$ and $\{\sigma_t\}_{t=1}^{T}$.

## 4. Experiments

**Experimental platforms.** We evaluate the effectiveness of our method in both real and simulated environments. For the real environment, we conduct experiments using the Cobot Magic robot (Agilex, 2024). As shown in Fig. 5, the robot

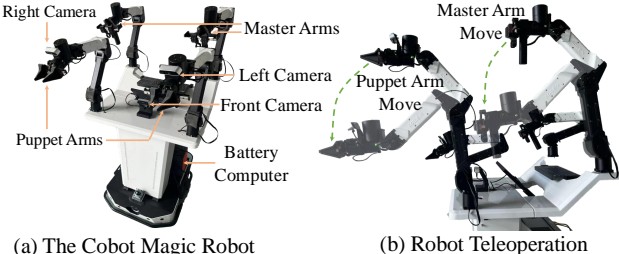

Figure 5. Illustrations of the Cobot Magic robot and how it is teleoperated. The robot has two master arms and two puppet arms.

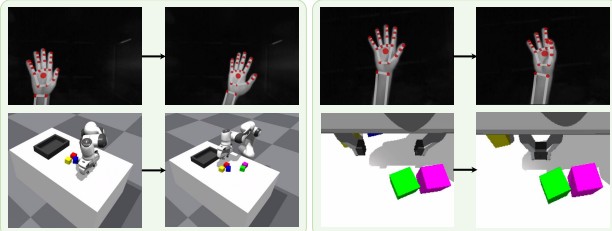

(a) Hand Pose based Arm Control  (b) Hand Pose based Gripper Control

Figure 6. Illustrations of how we teleoperate the robot in Isaac Gym. We build a real-time hand pose acquisition system to map the human hand pose to robot joint rotations. The orientation and translation of the palm are used to control the end of the robot arm. The distance between the thumb and index finger is employed to determine the opening or closing of the gripper.

is integrated with four robot arms, *i.e.*, two master arms and two puppet arms. When collecting demonstration data, we manually control the master arms, and the puppet arms imitate the actions of the master arms in real time. After fine-tuning the pre-trained policy with the demonstration data, the policy directly controls the puppet arms in inference. Three cameras are deployed on the robot, which are the right camera, front camera, and left camera, respectively.

The simulated environments are built based on Isaac Gym (Makoviychuk et al., 2021), which supports GPU-based efficient physics simulation. A Franka Panda robotic arm is deployed in each simulation environment to manipulate objects, with four cameras strategically positioned to observe the scene from various angles, including three peripheral views and one hand view. Unlike previous approaches that rely on manually crafted script rules for generating manipulation demonstrations (Zhao et al., 2023), we build a real-time hand pose acquisition system to teleoperate the simulated robotic arm, which better mimics the real demonstration data distribution. Specifically, a Leap Motion Controller (Ultraleap, 2013), which is a binocular infrared camera, is adopted to estimate the hand translation and orientation. Then, as shown in Fig. 6, we map the translation and orientation of the hand palm to the robot arm end-effector position using pre-defined rules, and the joint rotation angles of the robot arm are derived based on inverse kinematics (Kucuk & Bingul, 2006). The opening or closing

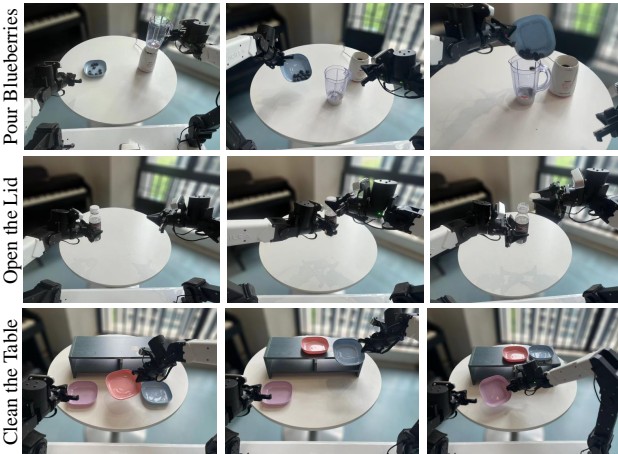

Figure 7. Illustrations of the three designed real-robot tasks, which include Pour Blueberries, Open the Lid, and Clean the Table.

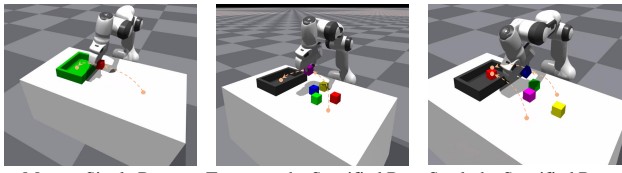

Move a Single Box  Transport the Specified Box  Stack the Specified Boxes

Figure 8. Illustrations of the three designed simulated tasks, which include Move a Single Box, Transport the Specified Box, and Stack the Specified Boxes.

of the robot gripper is controlled by the distance between the thumb and index finger of the human hand.

**Evaluation tasks**. To fully evaluate the effectiveness of our proposed techniques, we design three real-robot tasks and three simulated tasks. As depicted in Fig. 7, the three real-robot tasks include Pour Blueberries, Open the Lid, and Clean the Table. We collect 100 demonstrations of data for every task. In the Pour Blueberries task, the robot needs to first remove the juicer cup from the juicer and place it on the table. Then, the robot picks up the plate containing blueberries and pours all blueberries into the juicer cup. For the Open the Lid task, the robot uses a robotic hand to hold a bottle with a tightly screwed lid. The another hand first needs to grasp the lid. After a series of twists, the robot gradually unscrews and removes the lid from the bottle. In the Clean the Table task, three plates of different colors and a small cabinet are positioned on a table. The robot is required to move the plates onto the cabinet in a color order that is randomly specified during test. The three tasks test the multi-step operation, precise manipulation, and instruction following capabilities of policies, respectively.

The three simulated tasks are Move a Single Box, Transport the Specified Box, and Stack the Specified Boxes, as visualized in Fig. 8. We collect 50 demonstrations for the first task and 100 demonstrations for each of other two tasks. In Move a Single Box, the robot needs to transport the box on

*Table 1.* Effectiveness study of VIP in both real and simulated environments.

| Policy | Pre-train | Pour Blueberries ↑ | Open Lid ↑ | Clean Table ↑ | Move Box ↑ | Transport Box ↑ | Stack Boxes ↑ | Speed ↑ |
|---|---|---|---|---|---|---|---|---|
| ConvMLP | ✗ | 0.00 | 0.00 | 0.00 | 0.24 | 0.12 | 0.03 | 17.54 |
| ConvMLP | ✓ | 0.00 | 0.00 | 0.00 | 0.38 | 0.17 | 0.11 | 17.54 |
| Diffusion | ✗ | 0.19 | 0.46 | 0.24 | 0.72 | 0.51 | 0.43 | 27.32 |
| Diffusion | ✓ | 0.28 | 0.54 | 0.28 | 0.85 | 0.60 | 0.56 | 27.32 |
| ACT | ✗ | 0.28 | 0.51 | 0.25 | 0.84 | 0.61 | 0.47 | 43.48 |
| ACT | ✓ | 0.34 | 0.59 | 0.30 | 0.90 | 0.65 | 0.58 | 43.48 |
| VIRT (Ours) | ✗ | 0.30 | 0.65 | 0.29 | 0.87 | 0.65 | 0.54 | 39.22 |
| VIRT (Ours) | ✓ | 0.42 | 0.71 | 0.37 | 0.92 | 0.74 | 0.68 | 39.22 |

a table into a container. For Transport the Specified Box, five different colors of boxes are randomly located on a table, and the robot should move the box described by a random instruction to the container. Differently, in Stack the Specified Boxes, two boxes are specified. The robot is expected to move the first box in the container and then stack the second box on the first box.

**Implementation details**. In VIP, the pre-trained model parameters are updated using AdamW (Loshchilov, 2017) and the learning rate is $1e-5$. The action prediction horizon $T$ and image masking ratio $\tau$ are set to 20 and 0.5. Without a special statement, the cropped image is obtained from YOLOv10-small (Wang et al., 2024). We pre-train policies using Droid (Khazatsky et al., 2024) due to its large scale data volume and scene diversity. Besides, the manipulation trajectories in Droid present high ambiguity and are helpful for verifying the superiority of VIP. The pre-training consists of 120K iterations and fine-tuning comprises 8K iterations.

## 4.1. VIP Effectiveness

We study the effectiveness of VIP based on the aforementioned six tasks comprising Pour Blueberries, Open the Lid, Clean the Table, Move a Single Box, Transport the Specified Box, and Stack the Specified Boxes. Besides our developed VIRT, we further evaluate the effectiveness of pre-training on representative policies such as ConvMLP (Zhang et al., 2018), Diffusion Policy (Chi et al., 2023), and ACT (Zhao et al., 2023). Among them, ConvMLP is the most commonly adopted baseline, which first extracts image feature using convolutional neural network (CNN) and then regresses actions based on the extracted feature. Different from ConvMLP, Diffusion policy decodes the action chunk through iterative denoising. ACT consists of a CNN backbone, encoders, and decoders. Its basic architecture is similar to VIRT. For fair comparison, the cropped image region is input to all these policies as vision instruction. We pre-train these policies with the proposed pre-training paradigm described in Section 3 separately and then fine-tune them with the demonstration data of every task. These policies are tested for 100 times on each task, and we report their success rates as well as inference speeds (test on a

*Table 2.* Comparison among various instructions.

| Pre-train | Inference | Move Box | Transport Box | Stack Boxes |
|---|---|---|---|---|
| F | Cropped | 0.87 | 0.64 | 0.50 |
| S | Cropped | 0.78 | 0.51 | 0.36 |
| F+S | Text | 0.85 | 0.19 | 0.06 |
| F+S | Future | 0.88 | 0.67 | 0.54 |
| F+S | Cropped | 0.92 | 0.74 | 0.68 |

single RTX4090 GPU) in Table 1.

According to the results, we can observe that our designed pre-training paradigm is effective for all the policies in various model architectures. The success rates of these policies are boosted in both simulated and real robotic manipulation environments, indicating the value of incorporating more diverse training data.

Comparing the various policies, it is found that VIRT achieves the best performance, and its inference speed is also promising. For ConvMLP, its primary problem is its output head is a naive MLP, which is fast but fails to estimate actions precisely. In Diffusion Policy, since it adds random noise to action labels during training, the policy network is required to learn to recover actions from any noise. This requirement makes Diffusion Policy generally demand more data to converge well. VIRT is superior to ACT thanks to its broader perceptive field in encoders and the well-learned encoder representation inheriting from DINOv2.

## 4.2. Instruction Comparison

In this part, we analyze the effectiveness of different instructions used in pre-training and inference. The experiments are conducted based on the VIRT policy with the three simulated tasks. The experimental results are presented in Table 2. The studied pre-training instructions include only future image (F), only sparse point flows (S), and using both them (F+S). The analyzed inference instructions are text instruction, a world model predicted future image, and cropped images of target objects. According to the results, we can find that solely using a future image or sparse point flows does not lead to effective pre-training due to the lack of sufficient manipulation process description. By

Table 3. Ablation study on designs of VIRT.

| DINO | Uncern | Mask | Move Box | Transport Box | Stack Boxes |
|---|---|---|---|---|---|
| | | | 0.80 | 0.58 | 0.49 |
| ✓ | | | 0.86 | 0.66 | 0.57 |
| ✓ | ✓ | | 0.90 | 0.69 | 0.66 |
| ✓ | ✓ | ✓ | 0.92 | 0.74 | 0.68 |

Table 4. Study on the data scaling law.

| Pre-train | Fine-tune | Move Box | Transport Box | Stack Boxes |
|---|---|---|---|---|
| 0% | 100% | 0.87 | 0.65 | 0.54 |
| 10% | 100% | 0.81 | 0.55 | 0.49 |
| 50% | 100% | 0.89 | 0.68 | 0.60 |
| 100% | 10% | 0.40 | 0.25 | 0.06 |
| 100% | 50% | 0.77 | 0.58 | 0.43 |
| 100% | 100% | 0.92 | 0.74 | 0.68 |

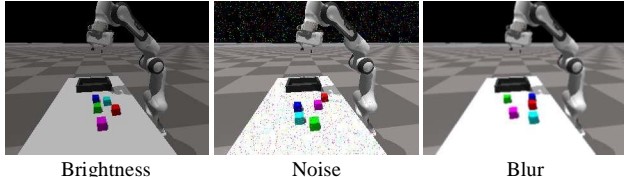

Brightness     Noise     Blur

Figure 9. Robustness analysis of VIRT to different disturbances, *e.g.*, brightness change, vision noise, and image blur.

Table 5. Analysis on the policy robustness.

| Brightness | Blur | Noise | Move Box | Transport Box | Stack Boxes |
|---|---|---|---|---|---|
| | | | 0.92 | 0.74 | 0.68 |
| ✓ | | | 0.86 | 0.67 | 0.60 |
| | ✓ | | 0.88 | 0.71 | 0.65 |
| | | ✓ | 0.75 | 0.64 | 0.52 |

contrast, combining them results in significant manipulation performance boosts, confirming the effectiveness of VIP.

Additionally, we can observe that although the policy based on text instruction obtains comparable performance with the ones using vision instructions on the Move a Single Box task, its performance on the other two tasks are much inferior. This is because there is only a single box on the table in Move a Single Box task, while the policy needs to grasp the correct box out of multiple boxes according to the provided instruction in the other two tasks. As visualized in Fig. 1, the text instruction based policy does not really understand the instruction content and often grasps an incorrect target. Moreover, it is found that the vision instruction based on a cropped image is superior than employing a future image. This is because of two reasons. On the one hand, the future image quality predicted by a world model is limited. On the other hand, the cropped image removes the disturbance from image background and guides the generated manipulation trajectories more clearly.

### 4.3. Method Analysis

**Ablation study**. This part conducts an ablation study on the designs in VIRT that are not clearly analyzed before. We mainly study the influences of three designs, *i.e.*, initializing encoder weight with DINOv2, uncertainty in supervision loss, and randomly masking the input images. Notably, to show the value of DINOv2 representation more clearly, when DINOv2 is not adopted, we replace the encoders in VIRT as ResNet18, the backbone that is widely adopted in previous robot policies. The experimental results are reported in Table 3. As shown, all these designs improve the success rates of VIRT on the three evaluated tasks significantly. Specifically, DINOv2 initialization provides the policy with the initial ability of extracting discriminative representation. Incorporating uncertainty into the supervision loss in Eq. (1) helps the policy concentrate on predicting the

trajectory parts that are less ambiguous. Ramdomly masking pixels of input images forces the Transformer-based policy to maintain its sensitivity to local features.

**Scaling law**. In this part, we study whether the data scaling law appears in the pre-training and fine-tuning procedures of robotic manipulation. Specifically, we conduct experiments using different amounts of the total pre-training and fine-tuning data, and the experimental results are presented in Table 4. According to the results, we can first observe that increasing both the pre-training and fine-tuning data is beneficial to improving the manipulation success rates. Interestingly, comparing the $1_{st}$ row (no pre-training is conducted) and $2_{nd}$ row of the results, it can be found that pre-training with only a little data harms the policy performance. This is because the policy is prone to over-fitting to the pre-training data domain if the pre-training data is insufficient, and there is a significant gap between the pre-training and fine-tuning data domains. In addition, we can find that increasing the fine-tuning data volume boosts execution success rates more significantly, which is because the fine-tuning data aligns better with inference observations.

**Policy robustness**. This part analyzes the robustness of VIRT to different unseen environment disturbances. As illustrated in Fig. 9, the studied disturbances include environment brightness, vision noise, and image blur. For environment brightness, we reduce the lightness intensity by 40%. To analyze the influence of random vision noise, we add random Gaussian noise to the perceived images. For image blur, we smooth the perceived images with a blur kernel with the shape of $3 \times 3$. We replace the input to the trained policy as the images with the studied noise and then test the success rates on various evaluation tasks. The experimental results are presented in Table 5. As shown, VIRT presents promising robustness to different environment changes thanks to the robust representation obtained from the large-scale pre-train stage. In addition, the Gaussian noise unseen in training data leads to the most

significant performance degradation.

## 5. Conclusion

In this work, we have revealed that vision instruction is more comprehensible than text instruction for current robotic policies. Based on this insight, we have designed VIP, which utilizes vision observations and sparse point flows to predict actions. To bridge the information gap between pre-training and applications, we have proposed to remove sparse point flows progressively during pre-training and replaced future images as cropped images of objects to manipulate. By combining the strengths of VIP with our developed policy VIRT, VIRT learns to complete diverse challenging tasks. Extensive experiments have been conducted in both real and simulated environments to confirm the effectiveness of our proposed techniques from various perspectives.

## Impact Statement

This paper presents work whose goal is to advance the field of robotic manipulation. There are many potential societal consequences of our work, none of which we feel must be specifically highlighted here.

## Acknowledgement

This work is supported by the National Natural Science Foundation of China (No. 62441615, 62422606, 62201484).

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
