# OpenReview forum: "VIP: Vision Instructed Pre-training for Robotic Manipulation"
_ICML.cc/2025/Conference — ICML 2025 poster_

### Official Review · Reviewer_DB9D · 2025-03-06

**Overall Recommendation:** 3

**Summary:**

The authors propose a pretraining strategy for robotic manipulation in which all inputs are visual, with no language descriptions required.

**Claims And Evidence:**

Yes

**Essential References Not Discussed:**

No

**Experimental Designs Or Analyses:**

Yes

**Methods And Evaluation Criteria:**

Yes

**Other Comments Or Suggestions:**

Additional baselines focusing on pretraining models for robotic tasks should be considered, such as R3M[3], VC-1[4] and others.

[3] Nair, Suraj, et al. "R3m: A universal visual representation for robot manipulation." arXiv preprint arXiv:2203.12601 (2022).

[4] Majumdar, Arjun, et al. "Where are we in the search for an artificial visual cortex for embodied intelligence?." Advances in Neural Information Processing Systems 36 (2023): 655-677.

**Other Strengths And Weaknesses:**

The paper is well-written, which is easy to follow. The real-world experiments are interesting, which include dual-arm setting.
However, my main concern lies in the motivation of this work, as described in the introduction, which highlights the drawbacks of language descriptions and emphasizes the importance of visual observations:

1. The paper uses RoboFlamingo to argue that language-guided policy models struggle to follow instructions due to data scarcity. However, I do not find this example convincing, as RoboFlamingo is not pretrained on a large-scale robotic dataset paired with language descriptions. Other works, such as OpenVLA[1] and PI0[2], pretrain models on large-scale robotic demonstrations with language descriptions, enabling instruction following—at least in simple cases like the 'green block' example provided by the authors. While I partially agree with the claim made in the second paragraph of the introduction, I do not believe that RoboFlamingo and the 'green block' example are appropriate choices to support this argument.

2. In the third paragraph of the introduction, the authors claim that visual prompts provide a more comprehensive representation for the model, leading them to use only visual information for pretraining, supported solely by a human intuition example. While I acknowledge that visual prompts have advantages over language prompts, their advantage should not be described as 'comprehensive to the model.' A visual model may be more comprehensive for a vision-motor policy, but this does not necessarily hold for a vision-language-action model.
Moreover, the intuition-based example does not sufficiently support the authors' argument. A citation to prior research would be more convincing.

3. If the claim that visual prompts provide a more comprehensive representation for the model is true, then how did the authors crop the image region of the object for manipulation during fine-tuning and inference without language guidance? If language is still used to guide the cropping process, then the claim that visual prompts alone are sufficient is inconsistent. The motivation should align with the expectation that the model fully leverages visual observations without relying on language descriptions.

[1] Kim, Moo Jin, et al. "Openvla: An open-source vision-language-action model." arXiv preprint arXiv:2406.09246 (2024).

[2] Black, Kevin, et al. "$\pi_0 $: A Vision-Language-Action Flow Model for General Robot Control." arXiv preprint arXiv:2410.24164 (2024).

**Questions For Authors:**

Refer to weakness section

**Relation To Broader Scientific Literature:**

Good

**Theoretical Claims:**

Yes

---

> ### Author Rebuttal · Authors · 2025-03-30
>
> We believe the Reviewer has significant misunderstandings of this work. In the following, we address the concerns one by one using more precise explanations and sufficient experiments. The paper will be revised accordingly to avoid these misunderstandings.
>
> ## Q1: Insufficient support for vision instruction
>
> We believe we do not mention RoboFlamingo in the paper. What we claim is in the current manipulation data status (very expensive to collect), vision instruction is more efficient for disclosing task information.
>
> We agree models like OpenVLA and PI0 can pick up the shown green box given text instruction We do not claim that text instruction cannot specify target. What we highlight is the efficiency difference between text and vision instructions. Robot manipulation data is expensive, and models like OpenVLA have used up most public data resources (Open X-Embodiment dataset covers most popular manipulation datasets). However, we find that OpenVLA overfits to specific text representation. For example, when we use the instruction ''grasp the cup", it works well. However, when we use ''pick up the cup'', the policy fails. This means even though we combine almost all public manipulation datasets, the data is still insufficient to cover common text instructions. By contrast, the shown generaliazation performance of our method is much better.
>
> To further support our claim, we compare the performances of the models pre-trained by OpenVLA, PI0, and our VIP. As the zero-shot manipulation results of OpenVLA and PI0 are limited, we tune them with the same data as our policy for fair comparison. Specifically, We use the pre-trained feature extractors to extract feature, and employ three Transformer decoders to decode the extracted feature as actions to execute. Each pre-trained policy is test in two settings, freeze or not freeze the feature extractor during tuning. Text instructions are provided to VLA-based policies. The success rates are as follows:
>
> Method | Freeze | Pour Blueberries | Open Lid | CLean Table | Move Box | Transport Box | Stack Boxes |
> | :-: | :-: | :-: | :-: | :-: |  :-: | :-: | :-: |
> PI0 | Yes | 0.07 | 0.48 | 0.14 | 0.82 | 0.44 | 0.29 |
> PI0 | No | 0.28 | 0.60 | 0.31 | 0.84 | 0.58 | 0.55 |
> OpenVLA | Yes | 0.14 | 0.45 | 0.19 | 0.81 | 0.45 | 0.36 |
> OpenVLA | No | 0.30 | 0.67 | 0.32 | 0.88 | 0.66 | 0.60 |
> VIP (Ours) | Yes | 0.32 | 0.63 | 0.34 | 0.88 | 0.69 | 0.61 |
> VIP (Ours) | No | 0.42 | 0.71 | 0.37 | 0.92 | 0.74 | 0.68 |
>
> According to the results, although PI0 and OpenVLA use much more pre-train data than VIP, VIP achieves better performance. This result supports our claim that vision instruction is more efficient than text instruction. The ablation study in Table 2 of the paper also supports this claim.
>
> In summary, our method achieves superior or at least similar performance as OpenVLA and PI0 based on much less data, which favors addressing the expensive data collection cost in robot manipulation.
>
> ## Q2: Vision instruction is more comprehensible
>
> In this work, "more comprehensible" means the policy can learn to follow instructions based on less training data. As reminded by the Reviewer, we will replace "comprehensible" as "easy to follow".
>
> We understand the concern of the Reviewer is VLA may can understand text instruction easily as it has been pre-trained using numerous image-text pairs. Actually, we have tried many VLAs during developing this work. The observation is large-scale image-text pre-train favors understanding text instruction in robot manipulation, but the required manipulation data volume required by text instruction is still significantly larger than vision instruction to learn to complete a task. This is due to the gaps between common image-text data and manipulation data. Image-text-action data is precious and far from being sufficient to train the policy to follow general text instruction following. The experiment results in the Reply to Q1 supports this observation.
>
> ## Q3: How the cropped region is obtained
>
> In test, we give a text instruction, and the text instruction is converted as vision instruction (cropped region) based on a 2D object detector.
>
> We do not claim we should not use text instruction. It is more convenient for human to give instructions based on text than taking a photo. What we claim is vision instruction is more efficient to serve as the direct input to the policy. Therefore, we convert the text instruction from human as vision instruction and input the vision instruction to the policy. The experiment results in Table 2 of the paper indicates this design leads to much better performance than inputting text instruction to the policy.
>
> ## Q4: Comparison with more pre-training methods
>
> As suggested by the Reviewer, we conduct more comparison with the recommended pre-train methods. Besides the results in the Reply to Q1, find more comparison results in the anonymous link: [Exp Result](https://anonymous.4open.science/r/VIP_rebuttal-9331/README.md).

---

> > ### Comment · Reviewer_DB9D · 2025-04-04
> >
> > In the first question, in the first paragraph, I was misled by an incorrect citation. Upon further checking, the reference that the paper cited was actually Visual Instruction Tuning, not RoboFlamingo, though the point I intended to make remains the same. However, the authors’ response effectively addressed my concern, so I’m willing to improve the score to Weak Accept.

---

### Official Review · Reviewer_Ee7z · 2025-03-10

**Overall Recommendation:** 4

**Summary:**

In this work, based on the observation that current policies cannot capture features from text instruction effectively, due to data scale, the authors propose a model named VIP which utilizes vision instructions to specify manipulation targets as an alternative. Specifically, the input of VIP is the current observation, future observation, and a corresponding sparse point flows between the current and future images. Since sparse point flows and future images are not available during inference time, they are progressively randomly masked during pre-training and future images are replaced with the cropped region of the object to manipulation in the current observation. The effectiveness of VIP is demonstrated with three tasks in the real world and three tasks in simulation(IsaacGym).

**Claims And Evidence:**

- This work claims that text-instructed policies cannot successfully capture the feature of manipulation targets because of the lack of sufficient manipulation data and image-text pairs. Figure 1, which displays the attention map of a text-instructed policy in a pick-and-place scenario, evidences this claim.

- The effectiveness of vision instructions is supported by the policy attention map in Figure 1 and the strong performance in both real and simulated environments.

**Essential References Not Discussed:**

Some works which pre-train on human demonstration datasets and are fine-tuned for robotic manipulation are not discussed in this work. These works involve but are not limited to:

[1] Ilija Radosavovic, Tete Xiao, Stephen James, Pieter Abbeel, Jitendra Malik, and Trevor Darrell. Real-world robot learning with masked visual pre-training. In CoRL, 2022.

[2] Suraj Nair, Aravind Rajeswaran, Vikash Kumar, Chelsea Finn, and Abhinav Gupta. R3M: A universal visual representation for robot manipulation. In CoRL, 2022.

[3] Jia Zeng, Qingwen Bu, Bangjun Wang, Wenke Xia, Li Chen, Hao Dong, Haoming Song, Dong Wang, Di Hu, Ping Luo, Heming Cui, Bin Zhao, Xuelong Li, Yu Qiao, Hongyang Li. Learning Manipulation by Predicting Interaction. In *RSS*, 2024.

[4] Arjun Majumdar, Karmesh Yadav, Sergio Arnaud, Yecheng Jason Ma, Claire Chen, Sneha Silwal, Aryan Jain, Vincent-Pierre Berges, Pieter Abbeel, Jitendra Malik, Dhruv Batra, Yixin Lin, Oleksandr Maksymets, Aravind Rajeswaran, Franziska Meier. Where are we in the search for an Artificial Visual Cortex for Embodied Intelligence? In arXiv, 2023.

Besides, there are also some works supporting utilizing both vision and text instructions, such as:

[5] Dibya Ghosh, Homer Walke, Karl Pertsch, Kevin Black, Oier Mees, Sudeep Dasari, Joey Hejna, Tobias Kreiman, Charles Xu, Jianlan Luo, You Liang Tan, Lawrence Yunliang Chen, Pannag Sanketi, Quan Vuong, Ted Xiao, Dorsa Sadigh, Chelsea Finn, Sergey Levine. Octo: An Open-Source Generalist Robot Policy. In RSS. 2024.

are not discussed in this work.

**Experimental Designs Or Analyses:**

It is promising that VIRT has achieved all state-of-the-art performance in the tasks given in the paper. However, those tasks involved are still very simple, pick-and-place tasks in a clean background, which cannot genuinely demonstrate the advantages of vision instructions over text instructions as proposed in the paper. It would be better if you could add more distractions such as more positional randomness of your manipulation targets, and more complex tasks, such as articulated object manipulation tasks, to your evaluation suite. The robustness to simple image transforms is not persuasive enough of its generalization capabilities and the effectiveness of the vision prompts.

**Methods And Evaluation Criteria:**

The proposed vision-instructed pre-training method makes sense for the manipulation tasks. The evaluations involving both tasks in the real world and the simulation are comprehensive, and the evaluation criteria are the success rates of each task.

**Other Comments Or Suggestions:**

It would be better if Table 1 could provide the parameter numbers of each method for direct comparison.

**Other Strengths And Weaknesses:**

The clarity of this work is great.

**Questions For Authors:**

Can VIRT also generalize to moving manipulation target scenarios?

**Relation To Broader Scientific Literature:**

Current pre-training methods in robotics manipulation mainly apply text as the prompt, while using vision instructions is less explored.

**Theoretical Claims:**

This work is mainly an ML application in robotic manipulation tasks, so there is no need to provide theoretical proofs.

---

> ### Author Rebuttal · Authors · 2025-03-30
>
> We have addressed the concerns of the Reviewer one by one in the following. The paper will be revised accordingly.
>
> ## Q1: Experiments with more distraction.
>
> As suggested by the Reviewer, we have added experiments with stronger distraction in both simulated and real robot environments. For simulation, we select a set of challenging tasks in RLbench, and these tasks require challenging actions like insert, open, and screw. We compare our method with previous SOTAs using the same training data and testing protocol. The results are reported as follows:
>
> Method | Insert Peg | Open Drawer | Place Cups | Put in Cupboard | Screw Bulb | Stack Cups |
> | :-: | :-: | :-: | :-: | :-: | :-: | :-: |
> ConvMLP | 0.00 | 0.04 | 0.00 | 0.00 | 0.00 | 0.00 |
> PolarNet | 0.04 | 0.84 | 0.00 | 0.12 | 0.44 | 0.08 |
> PerACT | 0.06 | 0.88 | 0.02 | 0.28 | 0.18 | 0.02 |
> ACT3D | 0.27 | 0.93 | 0.03 | 0.51 | 0.47 | 0.09 |
> RVT | 0.11 | 0.71 | 0.04 | 0.50 | 0.48 | 0.26 |
> RVT-2 | 0.40 | 0.74 | 0.38 | 0.66 | 0.88 | 0.69 |
> VIRT (Ours) | 0.48 | 0.79 | 0.42 | 0.68 | 0.90 | 0.73 |
>
> We can see that our method outperforms all previous methods.
>
> In addition,  we devise two new real robot tasks to verify the performance of our method. In both tasks, there are multiple targets randomly placed on a table. In the spoon deliver task, the robot hand needs to pick and deliver randomly placed spoons to a tray. This task is more challenging because spoons are smaller and the robot must learns to grasp the necks of the spoons in diverse poses. In the wire deliver task, the manipulation target is non-rigid and presents non-regular shapes. The policy must delivers all seen objects to achieve one successful trial. The success rates of different methods on these two new tasks are as follows:
>
> Method | CNNMLP | Diffusion | ACT | VIRT (Ours) |
> | :-: | :-: | :-: | :-: | :-: |
> Spoon Deliver | 0.00 | 0.06 | 0.14 |  0.42 |
> Wire Deliver | 0.00 | 0.00 | 0.07 | 0.39 |
>
> We can find that VIRT outperforms other methods more significantly on these more complex tasks. Moreover, we present anonymous video links of our method on these two tasks here: [Spoon Deliver](https://drive.google.com/file/d/1sEOaJ9PwOs8aV7beUxLOJdWNAVCEzFXL/view?usp=sharing) and [Wire Deliver](https://drive.google.com/file/d/111Jbot6Skb1uRL7u-a40oGxNglf4OYLo/view?usp=sharing).
>
> ## Q2: Method for Moving Object Manipulation.
>
> Yes, our method supports moving object manipulation due to its fast response speed and precise manipulation operation. We design a new task that the robot needs to pick the moving spoons on a moving belt and delivers them to a box. This task is challenging because (1) The spoons are small and in diverse poses. The robot hand must grasps the necks of spoons for successful picking up. (2) The spoons are moving, which means the policy must learns to predict the positions of spoons at the grasp moment rather than the current moment.
>
> The success rate of VIRT arrives 0.82, while all the other test methods fail. This result attributes to two advantages of our VIRT: (1) VIRT shows very fast response speed. (2) The vision instruction helps VIRT focus on the moving spoons and better predict the future positions of these spoons.
>
> We present an anonymous video link of our method on this new task here: [Move Manipulation](https://drive.google.com/file/d/111juTCA0EHkfemDUYPoUj0xdHg2Y6ldC/view?usp=sharing). To the best of our knowledge, this is the first end-to-end imitation policy that can perform moving object manipulation.
>
> ## Q3: More discussion with relevant works.
>
> We thank the Reviewer for this reminder. As suggested by the Reviewer, we will add all the references mentioned by the Reviewer and discuss the relevance of our work with them in the paper. The papers mentioned by the Reviewer are mostly about using video data cheaper than robot manipulation data (like human ego-centric videos and web videos) to conduct pre-train. The advantage of these works is the data is easy to obtain, but their effectiveness is limited due to embodiment gaps. Differently, what we study is another setting: how to use robot manipulation data to conduct pre-train, where the data is more expensive to collect but provides more direct generalization. Notably, these two settings of pre-train do not conflict with each other. We can first pre-train a policy using cheap web data and then fine-tune the policy based on the robot manipulation data based on our proposed method.
>
> ## Q4: Parameter number.
>
> As suggested by the Reviewer, we will report the parameter numbers of these methods in the paper. For the convenience of review, we also list them as follows:
>
> Method | CNNMLP | Diffusion | ACT | VIRT (Ours) |
> | :-: | :-: | :-: | :-: | :-: |
> Parameter Number | 76.7M  | 142.8M  | 115.0M  |  38.6M |
>
> According to the results, we can find that VIRT is the most lightweight model. This is mainly because VIRT adopts a very concise architecture consisting of only Transformer encoders and decoders.

---

> > ### Comment · Reviewer_Ee7z · 2025-04-04
> >
> > The authors' additional experiment results addressed my concerns about the method's capabilities of positional generalization and moving object manipulation. However, real-world experiments are still conducted in a clean and static environment without many distractions. So, I will accordingly increase my score to Accept.

---

### Official Review · Reviewer_HYVi · 2025-03-10

**Overall Recommendation:** 2

**Summary:**

This paper introduces a novel pretrained method designed for general robotic manipulation tasks. The authors argue that text-instructed policies often fail to effectively focus on target objects, and therefore, they propose integrating more interpretable features for pretraining visual-based policies. These features include sparse point flow and future visual observations, which enhance the model's ability to comprehend and predict environmental dynamics. To validate their approach, the authors have developed a series of tasks conducted in both real-world and simulated environments. These tasks are designed to rigorously evaluate the performance and robustness of the proposed method under various conditions.

**Claims And Evidence:**

The main claim regarding the drawback of text-instructed policies—that they may misinterpret the attention focus on the target object—is not sufficiently substantiated. While the authors provide a visualization in Figure 1 to compare text- and vision-instructed policies, two critical issues undermine the credibility of this comparison. First, the specifics of the text-instructed policy used for the visualization are not detailed, which raises questions about the validity and reproducibility of the results. Without transparency regarding the implementation and parameters of the text-instructed policy, it is difficult to assess the fairness of the comparison. Second, the claim appears to overlook the significant success and widespread adoption of text-instructed pretraining methods in the field. Text-instructed policies have demonstrated remarkable achievements across various pretraining tasks, making the assertion that they are inherently flawed seem overly subjective and potentially biased.

**Essential References Not Discussed:**

This paper introduces a novel pretraining paradigm; however, it overlooks one of the most widely recognized and influential pretraining methods in the field, like R3M, which was published at CoRL 2022.

**Experimental Designs Or Analyses:**

The soundness of the proposed method is not fully established. While the paper introduces a novel pretraining paradigm and, in Section 4, conducts experiments to validate the importance of individual components and the overall improvements for downstream tasks, it lacks a crucial comparison with other existing pretraining methods. This omission leads to the confusion about how the proposed method performs relative to previous pretraining methods. Without such comparisons, it is difficult to determine whether the claimed advancements are truly superior to state-of-the-art alternatives.

**Methods And Evaluation Criteria:**

Yes, integrating point cloud flows and future images into the pretraining framework could lead to more robust and generalizable policies.

**Other Comments Or Suggestions:**

1. Why not incorporate masks directly into the pretraining stage? Utilizing masks during pretraining could enhance consistency between the pretraining and inference phases. If the mask image is validated to be effective, it seems like the current pretraining process can also
rely on it instead of future images.

2. Is the pretraining model directly employed in the simulation environment? If so, the ability of the pretrained model should be avaluated in other imitation learning benchmark like RLBench or Robomimic.

**Other Strengths And Weaknesses:**

Strength:

1.The integration of additional visual features, such as future images and point flows, for pretraining a robust feature representation network is a notable strength. This approach enhances the performance of downstream action policies without relying on text embedding networks like CLIP, offering a more direct guidance for robotic manipulation tasks.

2.The authors conduct experiments in both real-world and simulated environments, which helps verify the practical effectiveness and versatility of the proposed method.

Weaknesses:

1.The claim regarding the drawbacks of text-instructed policies is not well-supported. Text-driven pretraining paradigms, such as R3M, are widely adopted in the field and have demonstrated strong generalizability in robotics. The lack of a fair comparison or evidence to substantiate this claim undermines its validity.

2.A significant limitation is the absence of experiments comparing the proposed method with other established pretraining approaches, such as CLIP-based or R3M-based methods. This omission weakens the confidence in the superiority or even the competitiveness of the proposed method against existing alternatives.

3.The presentation of the algorithm's details is insufficient. While the paper mentions the integration of images and point flows during training, it fails to provide a clear explanation of how these features are processed or combined, leaving critical implementation aspects unclear.

4.Although the authors propose a general pretraining method, they do not evaluate it on widely used benchmarks like RLBench or Robomimic. Such evaluations are essential to demonstrate the generalizability and robustness of the proposed method across diverse tasks and environments, which is a key expectation for a general pretraining framework.

**Questions For Authors:**

Please refer to the comments_suggestions and the strenghs_weaknesses sections.

**Relation To Broader Scientific Literature:**

The authors propose a novel pretraining method for robotic manipulation tasks. Unlike previous text-driven pretraining methods such as R3M and CLIP, this work shifts the focus to a vision-based paradigm. Text-driven methods typically align given prompts with current visual observations to generate conditional features for downstream policies. The authors identify potential limitations in text-instructed policies, like
misinterpretations or illusions in aligning textual instructions with visual contexts. To address this, they design a novel vision-based pretraining framework. A key contribution of the proposed method lies in its explicit use of future images and point cloud flows, which clearly indicate the intended direction of target actions. By focusing on visual cues rather than relying on potentially ambiguous textual instructions, the proposed method could be able to overcome some of the inherent limitations of text-driven approaches.

**Theoretical Claims:**

This paper contains no theoretical claims.

---

> ### Author Rebuttal · Authors · 2025-03-30
>
> We have addressed the concerns of the Reviewer one by one in the following. The paper will be revised accordingly.
>
> ## Q1: Visualization details of the teaser figure.
>
> In the teaser figure, we visualize the attention map between the action query $q_T$ at the last timestamp $T$ and image feature tokens $F$ in the last decoder. We use $q_T$ as query and $F$ as key to compute the attention map $A$. Then, we take the mean of $A$ across all channels and resize the obtained feature to the same shape of the input image. Finally, we visualize the feature and get the teaser figure. For fair comparison, the text and vision instructed policies adopt the same network except the input instructions are different. The policy parameter number is 38.6M.
>
> ## Q2: The success of previous text-instructed methods.
>
> * The mentioned pre-train methods like R3M are based on natural videos (like human hand operation videos). Although these models provide promising initialization weight for robot applications, their effectiveness is limited due to embodiment and task gaps. Differently, what we study is another setting: how to use robot manipulation data to conduct pre-train, where the data is more expensive to collect but provides more direct generalization.
>
> * There are text-instructed VLAs based on robot manipulation data, like OpenVLA. However, we find these methods overfit to specific text instructions. For example, OpenVLA behaves well given ''grasp the cup'' but fails given ``pick up the cup''. This is because manipulation data is expensive to collect and not diverse enough to cover common text instructions, although the works like OpenVLA have used up most public data resources.
>
> * We have added comparison experiments with previous pre-train methods. For fair comparison, we integrate their pre-trained backbone into VIRT in the same way as our VIP pre-trained backbone. If the pre-train method relies on text instruction, we provide the policy with text instruction during fine-tune and test. We report comparison results using all the simulated and real-robot tasks as follows:
>
> Method | Pour Blueberries | Open Lid | Clean Table | Move Box | Transport Box | Stack Boxes |
> | :-: | :-: | :-: | :-: | :-: | :-: | :-: |
> R3M | 0.26 | 0.62 | 0.25 | 0.83 | 0.68 | 0.51 |
> OpenVLA | 0.30 | 0.67 | 0.35 | 0.88 | 0.66 | 0.60 |
> E-RADIO | 0.34 | 0.68 | 0.32 | 0.84 | 0.69 | 0.62 |
> Theia-B | 0.38 | 0.65 | 0.36 | 0.85 | 0.65 | 0.64 |
> VIP (Ours) | 0.42 | 0.71 | 0.37 | 0.92 | 0.74 | 0.68 |
>
> According to the results, we can find that although the compared methods generally use more pre-train data than ours, our proposed method achieves better performance.
>
> ## Q3: Comparison with other pre-train methods.
>
> As suggested by the Reviewer, we have conducted more comparison between our pre-train method and other pre-train methods. Due to character limit, the results are reported in the above Table in the Reply to Q2. The results suggest that our pre-train method achieves the best performance on all tasks, which further reveals the superiority of our method.
>
> ## Q4: Comparison on public benchmarks.
>
> As suggested by the Reviewer, we compare our method with previous SOTAs using the ManiSkill-v3 and RLBench benchmarks. The training data and testing protocols are the same between VIRT and the compared methods. Refer to the reply to Q2 of Reviewer uDoz for more experiment details.
>
> The results on ManiSkill-v3 are as follows:
>
> Method | Push Cube | Pick Cube | Stack Cube | Draw Triangle |
> | :-: | :-: | :-: | :-: | :-: |
> ConvMLP | 0.32 | 0.56 | 0.00 | 0.02
> Diffusion | 0.75 | 0.81 | 0.06 | 0.55
> ACT | 0.84 | 0.89 | 0.20 | 0.68 |
> VIRT (Ours) | 0.94 | 0.96 | 0.48 | 0.80 |
>
> For RLBench, we use the training data of PerACT and follow its testing protocol. The results on RLBench are as follows:
>
> Method | Insert Peg | Open Drawer | Place Cups | Put in Cupboard | Screw Bulb | Stack Cups |
> | :-: | :-: | :-: | :-: | :-: | :-: | :-: |
> ConvMLP | 0.00 | 0.04 | 0.00 | 0.00 | 0.00 | 0.00 |
> PolarNet | 0.04 | 0.84 | 0.00 | 0.12 | 0.44 | 0.08 |
> PerACT | 0.06 | 0.88 | 0.02 | 0.28 | 0.18 | 0.02 |
> ACT3D | 0.27 | 0.93 | 0.03 | 0.51 | 0.47 | 0.09 |
> RVT | 0.11 | 0.71 | 0.04 | 0.50 | 0.48 | 0.26 |
> RVT-2 | 0.40 | 0.74 | 0.38 | 0.66 | 0.88 | 0.69 |
> VIRT (Ours) | 0.48 | 0.79 | 0.42 | 0.68 | 0.90 | 0.73 |
>
> The results on the ManiSkill and RLBench benchmarks show VIRT achieves the best results, which further confirm the superiority of our method.
>
> ## Q5: How image and point flow are integrated.
>
> As shown in the pipeline of the paper, the images and points are encoded as tokens separately. Then, we directly concatenate them as the feature input to Transformer decoders. We will release our code publicly to help other researchers reproduce our work.
>
> ## Q6: Image random mask in pre-train.
>
> The image random mask improves the robustness of the pre-train policy but cannot directly enable the policy to learn useful representation. Therefore, the future action prediction task is necessary in pre-train.

---

### Official Review · Reviewer_uDoz · 2025-03-13

**Overall Recommendation:** 4

**Summary:**

This paper proposes VIP, a framework that uses visual signals as guidance for visual pre-training in robotic tasks. Instead of using language as the instruction (which is ambiguous), this work proposes using vision-instructed policy as an alternative. The vision-instructed policy uses visual cues to instruct the policy on what to do and where to pay attention, which helps policy learning. During training, the visual signals (sparse points) are masked and annealed gradually, which helps policy training by making the sparse points unnecessary at inference time. Experiments show that the policy is able to boost performance in both simulated and real-world robots.

**Claims And Evidence:**

This paper claims that using vision-instructed pre-training provides a performance boost to downstream tasks and proposes using target image + sparse key point as an effective pre-training strategy for simulated and robotic bi-manual manipulation tasks. The evidence in performance boosts backs up these claims.

**Essential References Not Discussed:**

Reference is adequately discussed.

**Experimental Designs Or Analyses:**

The experiment design and analysis are adequate to back up the claims. I appreciate the extensive ablations to show each component of the proposed method. Based on the success rate, it is clear that the proposed pre-training method is effective in boosting the performance of the policy.

**Methods And Evaluation Criteria:**

Three real-world and simulated robotic tasks are used to evaluate the proposed method. These tasks are household tasks that can evaluate the performance of the visual policy. There is a missing opportunity in evaluating in standard robotic benchmarks such as ManiSkill or RobotSuite.

**Other Comments Or Suggestions:**

I suggest testing the proposed method on standard benchmarks to better compare it with SOTA methods.

**Other Strengths And Weaknesses:**

### Strength
- I find the idea of using visual signals and sparse key points as guidance an intuitive and useful idea for visual pretraining. Figure 1 shows the benefit of using vision-instructed policy well. The sparse point flow is an interesting idea to explore as a different modality to instruct policy learning.
- Extensive ablations show that the combination of vision instruction and sparse keypoints achieves the best performance.
- SOTA results achieved compared to baselines.
- Robustness analysis shows that the policy learned with pre-training is robust to visual noises.

### Weakness
- Most of the ablations are conducted in simulation but not in the real world. Real-world experiments can lead to different results than simulation.

**Questions For Authors:**

None

**Relation To Broader Scientific Literature:**

This work tackles the important question of how to use large-scale pre-training data for training visual encoders for robotic tasks. Due to the embodiment difference, it is important to train visual feature extractors that can extract useful information from the existing datasets.

**Theoretical Claims:**

No theoretical claims made.

---

> ### Author Rebuttal · Authors · 2025-03-30
>
> We have addressed the concerns of the Reviewer one by one in the following. The paper will be revised accordingly.
>
> ## Q1: Ablation study in real-world experiments.
>
> In the paper, we conduct ablation study mainly in simulation as the environment is easy to control, which benefits to ensuring fair comparison. Our experience shows that the experimental results between our built simulation environment and real robot are consistent.
>
> As suggested by the Reviewer, we also conduct ablation study using the real robot and report the success rates on all the three our designed real-robot tasks. These three tasks evaluate the performance of robot manipulation policies from different perspectives.
>
>
> The real-robot experiment results corresponding to Table 2 of the paper are as follows. We can see that the improvments of our proposed designs are more significant in the real-robot experiments.
>
> Pre-train | Inference | Pour Blueberries | Open Lid | Clean Table |
> | :-: | :-: | :-: | :-: | :-: |
> F | Cropped | 0.37 | 0.63 | 0.30 |
> S | Cropped | 0.00 | 0.02 | 0.00 |
> F+S | Text | 0.37 | 0.63 | 0.14 |
> F+S | Future | 0.30 | 0.57 | 0.06 |
> F+S | Cropped | 0.42 | 0.71 | 0.37 |
>
> The real-robot experiment results corresponding to Table 3 of the paper are as follows:
>
> DINO | Uncern | Mask | Pour Blueberries | Open Lid | Clean Table |
> | :-: | :-: | :-: | :-: | :-: | :-: |
> No | No | No | 0.29 | 0.56 | 0.26 |
> Yes | No | No | 0.33 | 0.63 | 0.31 |
> Yes | Yes | No | 0.39 | 0.65 | 0.34 |
> Yes | Yes | Yes | 0.42 | 0.71 | 0.37 |
>
> According to the reported results, all the designs in our proposed method are also effective on the real robot.
>
> ## Q2: Experiment on public simulator benchmarks.
>
> In the paper, we conduct experiments based on our own built simulation environment rather than previous public benchmarks because these public benchmarks usually generate demonstration data by rule-based motion control. Their generated data has a distribution gap with the real robot data collected based on teleoperation. Differently, our used training data in our own built simulation tasks is also obtained from teleoperation, so the experimental results are more convinicing.
>
> As suggested by the Reviewer, we compare our method with previous SOTAs using the ManiSkill-v3 and RLBench benchmarks. We select them because they provide unified training data generation protocol, which ensures fair comparison. The training data and testing protocols are the same between VIRT and the compared methods.
>
> The tasks in ManiSkill-v3 cover navigation, humanoid robot, manipulation, etc. We select the manipulation tasks in ManiSkill-v3 to conduct experiments. As mentioned before, the training data is generated based on classic motion control.  The results on ManiSkill-v3 are as follows:
>
> Method | Push Cube | Pick Cube | Stack Cube | Draw Triangle |
> | :-: | :-: | :-: | :-: | :-: |
> ConvMLP | 0.32 | 0.56 | 0.00 | 0.02
> Diffusion | 0.75 | 0.81 | 0.06 | 0.55
> ACT | 0.84 | 0.89 | 0.20 | 0.68 |
> VIRT (Ours) | 0.94 | 0.96 | 0.48 | 0.80 |
>
> For RLBench, although there are totally 100 tasks, many tasks are easy and previous methods have achieved very high success rates. Due to the reply character limit, we select six challenging and representative tasks that the success rate of the best method is still below 90%. We compare our method with the other methods that have reported their results in RLbench. We use the training data of PerACT and follow its testing protocol (the other compared methods also adopt this protocol). The results on RLBench are as follows:
>
> Method | Insert Peg | Open Drawer | Place Cups | Put in Cupboard | Screw Bulb | Stack Cups |
> | :-: | :-: | :-: | :-: | :-: | :-: | :-: |
> ConvMLP | 0.00 | 0.04 | 0.00 | 0.00 | 0.00 | 0.00 |
> PolarNet | 0.04 | 0.84 | 0.00 | 0.12 | 0.44 | 0.08 |
> PerACT | 0.06 | 0.88 | 0.02 | 0.28 | 0.18 | 0.02 |
> ACT3D | 0.27 | 0.93 | 0.03 | 0.51 | 0.47 | 0.09 |
> RVT | 0.11 | 0.71 | 0.04 | 0.50 | 0.48 | 0.26 |
> RVT-2 | 0.40 | 0.74 | 0.38 | 0.66 | 0.88 | 0.69 |
> VIRT (Ours) | 0.48 | 0.79 | 0.42 | 0.68 | 0.90 | 0.73 |
>
> The results on the ManiSkill-v3 and RLBench benchmarks show VIRT achieves the best results, which further confirm the superiority of our method.

---

> > ### Comment · Reviewer_uDoz · 2025-04-06
> >
> > Thanks for the additional results! My concerns are addressed and I will maintain my acceptance rating.

---

### Decision · Program_Chairs · 2025-05-01

**Decision:**

Accept (poster)

**Comment:**

This paper proposes VIP, a vision-instructed pretraining framework for robotic manipulation that uses visual cues as instruction, instead of relying on potentially ambiguous language inputs. The key idea is to guide policy learning using purely visual signals, allowing robust pretraining while discarding the need for language-based annotation or alignment. The authors conduct extensive evaluations on both simulated and real-world tasks and demonstrate improvements over existing baselines.

Reviewer uDoz recommends accept, acknowleging the novelty of using visual signals as instruction and the strong empirical results, especially in real-robot settings. The reviewer’s concerns about real-world ablations were addressed by new experiments added in the rebuttal. Reviewer Ee7z also recommends accept after the rebuttal. Initially concerned with the simplicity of tasks, this reviewer was satisfied by the newly added results on more complex and dynamic environments (e.g., moving object manipulation and multi-target settings). Reviewer DB9D adjusts their rating to weak accept, for the additional clarifications around the motivation and claims about vision-vs-language instruction efficiency. The authors provided concrete comparison results against PI0 and OpenVLA, demonstrating better or comparable performance despite using less training data.

Reviewer HYVi remains at weak reject, maintaining that the drawbacks of text-instructed policies were not sufficiently justified and that certain implementation and comparison details were still lacking. Nevertheless, many of the key concerns were addressed in the rebuttal, including new experiments on RLBench and ManiSkill, network details, and expanded comparisons.

Given the overall positive consensus, the detailed and data-rich rebuttal, and the importance of the problem setting, I recommend acceptance.